# Relaxation and Devitrification of Mg_66_Zn_30_Ca_4_ Metallic Glass

**DOI:** 10.3390/ma18112464

**Published:** 2025-05-24

**Authors:** Karel Saksl, Juraj Ďurišin, Martin Fujda, Zuzana Molčanová, Beáta Ballóková, Miloš Matvija, Katarína Gáborová

**Affiliations:** 1Institute of Materials, Faculty of Materials, Metallurgy and Recycling, Technical University of Košice, Letná 1/9, 042 00 Košice, Slovakia; martin.fujda@tuke.sk (M.F.); milos.matvija@tuke.sk (M.M.); katarina.gaborova@tuke.sk (K.G.); 2Institute of Materials Research of SAS, Slovak Academy of Sciences, Watsonova 47, 040 01 Košice, Slovakia; molcanova@saske.sk (Z.M.); bballokova@saske.sk (B.B.); 3Department of Technologies in Electronics, Faculty of Electrical Engineering and Informatics, Technical University of Košice, Letná 9, 042 00 Košice, Slovakia; juraj.durisin@tuke.sk

**Keywords:** biodegradable alloys, metallic glasses, relaxation, devitrification, phase composition

## Abstract

Mg_66_Zn_30_Ca_4_ metallic glass is a promising biodegradable material due to its high strength, corrosion resistance, and excellent glass-forming ability. In this study, we investigated its thermal stability, structural relaxation, and crystallization behavior using high-energy synchrotron-based X-ray diffraction and DSC analysis. The glass exhibits a wide supercooled liquid region of 58 K, allowing for thermoplastic forming. Structural relaxation experiments revealed nearly a complete relaxation in the first cycle below the first crystallization peak. Upon heating, the alloy undergoes a complex, multi-step devitrification involving successive formation of crystalline phases: Mg_51_Zn_20_ (orthorhombic), Mg (hexagonal), and a Ca–Mg–Zn intermetallic compound Ca_8_Mg_26.1_Zn_57.9_, denoted as *IM3*. Phase identification was carried out by Rietveld refinement, and the evolution of lattice parameters demonstrated anisotropic thermal expansion, particularly in the Mg_51_Zn_20_ phase. Notably, the presence of the *IM1* Ca_3_Mg_x_Zn_15−x_, with the 4.6 ≤ *x* ≤ 12 phase reported in earlier studies, was not confirmed. This work deepens the understanding of phase stability and crystallization mechanisms in Mg-based metallic glasses and supports their future application in biodegradable implants.

## 1. Introduction

In recent years, Mg-based metallic glasses have emerged as a class of promising materials due to their unique combination of physical and mechanical properties. Unlike their crystalline counterparts, these amorphous alloys exhibit high yield strength, extended elastic strain limits, and superior corrosion resistance—attributes primarily derived from their dislocation-free atomic structure [1,2,3,4,5,6]. Their low density (<3 g·cm^−3^) and favorable biocompatibility render them particularly attractive for biomedical applications, especially in the design of temporary osteosynthesis implants where degradability, mechanical compatibility with bone, and bioresorption are essential criteria [7,8,9,10]. The addition of alloying elements such as calcium (Ca) and zinc (Zn) further improves mechanical strength, creep resistance, and corrosion behavior [11,12,13].

Among these systems, the Mg–Zn–Ca ternary alloys have attracted considerable attention due to their excellent glass-forming ability (GFA) and potential as bioresorbable materials [14,15]. In particular, Mg_66_Zn_30_Ca_4_ metallic glass is known for its high critical casting thickness (~5 mm), exceptional compressive strength (716–854 MPa), and an elastic modulus (31 GPa) that closely approaches that of human cortical bone. This alloy also exhibits minimal hydrogen evolution during degradation, making it suitable for medical use [8].

In our earlier work [16], we characterized the atomic structure of Mg_66_Zn_30_Ca_4_ glass and demonstrated the presence of a high fraction of rigid, icosahedrally packed clusters. These topological features inhibit atomic diffusion, delay crystallization, and expand the supercooled liquid region—conditions favorable for thermoplastic processing.

Despite significant advances, the thermal stability and phase evolution during devitrification of this glass remain incompletely understood. Previous studies have identified multiple crystallization events during continuous heating, but the exact nature and sequence of phase formation remain controversial and partly unresolved [17,18]. It is particularly important to identify the primary crystallization products and intermediate phases, as these influence both the mechanical integrity and degradation behavior of the alloy.

This study aims to provide a comprehensive characterization of the relaxation and crystallization behavior of Mg_66_Zn_30_Ca_4_ metallic glass. Using high-energy synchrotron-based X-ray diffraction in transmission mode, we examined the structural evolution of the alloy during thermal cycling and continuous heating beyond its melting point. In particular, we focused on the early stages of relaxation, identified phase transitions during devitrification, and refined the lattice parameters and thermal expansion of the resulting crystalline phases. These results contribute to a better understanding of phase stability in bioresorbable metallic glasses and offer insights for future materials design in structural and biomedical applications.

## 2. Materials and Methods

The Mg–Zn–Ca alloy with a nominal composition of Mg_66_Zn_30_Ca_4_ (at.%) was synthesized by induction melting of high-purity elements (Ca: 99.5 wt.%, Mg: 99.98 wt.%, Zn: 99.99 wt.%) in a boron nitride (BN)-coated quartz tube under high vacuum conditions (less than 3.0 × 10^−3^ Pa) at 1000 °C. Amorphous ribbons were subsequently fabricated from the melt by the single-roller melt-spinning technique, resulting in specimens with a maximum thickness of ~40 μm, width of approximately 5 mm, and a length under 100 mm. The chemical composition of the resulting amorphous ribbon was verified using EDX analysis in the Jeol JSM 7000F - JEOL Ltd., Tokyo, Japan scanning electron microscope. The measured composition was Mg = 65.7 at.%, Zn = 29.8 at.%, and Ca = 4.5 at.%, which is in very good agreement with the nominal target composition and indicates only minimal evaporation during processing.

Thermal analysis was conducted using a power-compensated differential scanning calorimeter (DSC 8000, Perkin Elmer—Waltham, MA, USA). Continuous heating experiments were performed at a scan rate of 5 K·min^−1^ in a platinum pan under an argon atmosphere. Calibration of temperature and enthalpy was performed using pure indium. Each thermal scan was followed by a second run to obtain the baseline.

High-energy X-ray diffraction (XRD) experiments were carried out in transmission (Debye–Scherrer) geometry using the experimental hutch P02.1 Powder Diffraction and Total Scattering undulator Beamline [19] at the PETRA III synchrotron source (6.0 GeV, 100 mA, operated in top-up mode) [20]. The experimental setup involved a monochromatized X-ray beam with a photon energy of ~60 keV (λ = 0.021014 nm). The beam had a cross-section of ~0.5 mm × 0.5 mm. A fast 2D Perkin Elmer XRD 1621 detector—Waltham, MA, USA (2048 × 2048 pixels, pixel size: 200 μm × 200 μm) recorded the diffraction patterns [21]. Ribbon specimens were cut into small pieces and loaded into quartz capillaries (inner diameter: 0.8 mm), which were then placed inside a Linkam THMS600 resistive heating stage (Linkam Scientific Instruments Ltd., Tadworth, Surrey, UK) with a thermal stability better than 1 K [22].

During the experiment, two types of XRD measurements were conducted. The first, aimed at investigating the relaxation behavior of the metallic glass, employed the detector positioned close to the sample (311 mm), enabling the collection of scattering data up to a maximum scattering vector of *Q_max_ = 4πsin(θ)/λ* = 150 nm^−1^. Under this setup, the sample was heated at a rate of 5 K·min^−1^ from room temperature to 396 K, which is just below the onset of the first crystallization event in this metallic glass. After reaching this temperature, the sample was cooled back to room temperature at the same rate. This heating–cooling cycle was repeated twice, and, throughout both cycles, 2D XRD patterns were collected every 2 K.

Following this measurement, the detector was moved to a further position (~1002 mm, corresponding to *Q_max_* = 82 nm^−1^), allowing for the acquisition of diffraction data with higher angular resolution. The second study focused on tracking the crystallization process of the amorphous Mg_66_Zn_30_Ca_4_ alloy. The same sample was continuously heated from room temperature to 753 K (approximately 100 K above its melting point) at the same heating rate of 5 K·min^−1^. As in the relaxation experiment, 2D XRD patterns were recorded every 2 K.

All 2D data were subsequently integrated into the *I(2θ)* space using the Fit2D (version V18.002) software [23]. However, the data evaluation approaches differed between the two experiments. For the relaxation study, X-ray structure factors *S(Q)* were calculated from the integrated data using the Faber–Ziman formalism [24] implemented in the PdfGetX2 program [25]. The mathematical formulation and processing details can be found in our previous work [26]. The second experiment focused on the devitrification process of the Mg_66_Zn_30_Ca_4_ metallic glass, followed standard crystallographic procedures—beginning with phase identification followed by Rietveld refinement of the identified phase models according to the measured experimental data. This evaluation was performed using the GSAS-II software package (version 5304) [27].

## 3. Results and Discussion

### 3.1. Thermal Stability of the Mg_66_Zn_30_Ca_4_ Metallic Glass

Figure 1 shows the differential scanning calorimetry (DSC) curve illustrating the thermal response of the amorphous Mg_66_Zn_30_Ca_4_ metallic alloy. Upon continuous heating from room temperature to the molten state, the alloy undergoes several phase transformations, as indicated in the insert of the figure. The glass transition temperature (*T_g_*) of the alloy was observed at 342 K, while the onset of the first crystallization (*T_x_*_1_) occurred at 400 K. The temperature interval between these two points (58 K), known as the supercooled liquid region, is characterized by a rapid drop in viscosity, which allows easy thermoplastic forming of the metallic glass within this temperature window.

For example, in the case of Mg_65_Cu_25_Y_10_ metallic glass, a transition through the glass transition temperature into the superplastic regime reduced its viscosity from 10^11^ Pa·s to 10^−3^ Pa·s [28]. Generally, the broader this temperature window is, the better is the formability of the amorphous material. The onset of crystallization in the amorphous precursor is accompanied by release of internal energy (exothermic effect), a sharp increase in viscosity, and a transition of the material into a brittle state.

The high thermal stability of this metallic glass, reflected in the elevated glass transition and delayed crystallization, is attributed to its short-range atomic structure. Previous studies [16] have shown a dominant presence of icosahedral clusters in Mg_66_Zn_30_Ca_4_, which reduce atomic mobility and act as kinetic barriers to phase transformation.

The first crystallization peak (*T**x*_1_) appears at 409 K, the second (*T**x*_2_) at 470 K, the third (*T**x*_3_) at 504 K, and it is likely that the fourth (*T**x*_4_) occurs near the melting point, around 612 K. The onset of melting (endothermic transition) is observed at 622 K, with the melting point at 634 K, and the liquidus temperature (*T*_*L*_)—at which the sample is fully molten—at approximately 643 K. In general, these transitions indicate a complex, multi-phase crystallization process triggered by thermal exposure, involving the formation of several distinct crystalline phases.

### 3.2. Relaxation of the Mg_66_Zn_30_Ca_4_ Metallic Glass

As described in the Section 2, the first diffraction experiment focused on evaluating the structural relaxation of the metallic glass in the temperature range from room temperature up to the onset of the first crystallization event. The amorphous Mg_66_Zn_30_Ca_4_ alloy was subjected to two complete heating–cooling cycles up to 396 K at a rate of 5 K·min^−1^. Diffraction patterns were recorded approximately every 2 K, from which the X-ray structure factor, *S*(*Q*), was extracted. Figure 2a displays the structure factor, *S*(*Q*), of the as-quenched (melt-spun) sample in red, and that of the alloy after two heating–cooling cycles in black. As shown, the differences between the two structure factors are minimal. A key finding, however, is that, under the applied heating and cooling conditions, the amorphous sample exhibits no signs of crystallization.

The degree of structural relaxation in the metallic glass can be quantified by the integrated difference in the structure factors, specifically$$\sum \left|{S(Q)}_{relaxed}-{S(Q)}_{T}\right|$$ , where *S*(*Q*)_*T*_ is the structure factor at temperature *T*, and *S*(*Q*)*_relaxed_* is the structure factor of the relaxed state. For this analysis, the relaxed reference state was chosen to be *S(Q)*, recorded at 300 K after the second cooling cycle.

From this comparison, it can be concluded that the alloy structure undergoes nearly complete relaxation during the first heating cycle. Further thermal cycling has only a minor effect on relaxation. The relaxation process involves a rearrangement of the amorphous phase, allowing more efficient filling of free volume while maintaining an apparently random structure. This leads to a reduction in configurational entropy and drives the system into a more energetically favorable state. It is worth noting that the entropy changes in the system correlate directly with the introduced metric of structural difference. Moreover, it is evident from Figure 2 that the most significant changes in structure during the first heating cycle occur from the glass transition temperature, *T*_*g*_, i.e., within the supercooled liquid region.

### 3.3. Crystallization of the Mg_66_Zn_30_Ca_4_ Metallic Glass

The second experiment was dedicated to studying the devitrification process of the Mg_66_Zn_30_Ca_4_ metallic glass. After modifying the experimental setup, the previously relaxed sample was heated from room temperature to above its melting point. The results of this measurement are shown in the 3D plot in Figure 3a, which displays the evolution of the structure factors at different annealing temperatures. A top-down view of the same dataset is presented in Figure 3b.

As seen in Figure 3a,b, the alloy undergoes several phase transformations during heating. Up to 409 K, the sample remains in the glassy state, as evidenced by the broad, low-intensity oscillations in *S*(*Q*), which are characteristic for amorphous materials. Beyond this temperature, the shape of *S*(*Q*) changes significantly, and Bragg peaks begin to emerge on top of the amorphous background—indicative of the formation of ordered crystalline phases.

Further abrupt changes in the diffraction patterns are observed at approximately 470 K, 504 K, and 575 K, indicating successive phase transitions. The final transformation corresponds to the melting of the alloy at around 643 K, after which the diffraction pattern once again resembles that of an amorphous material.

In our previous work [29,30], we introduced a method to quantify structural changes by calculating the difference in the area under the structure factor, *S*(*Q*), at temperature *T* and that of the preceding scan at *T*−2K. This metric, referred to as the instantaneous structural change, is defined as:$$\sum \left|{S(Q)}_{T}-{S(Q)}_{T-2K}\right|$$

Figure 3c,d compare this instantaneous structural change signal with the DSC trace. Both signals show transitions at the same characteristic temperatures. Notably, a pronounced peak is observed at 575 K in the structural signal (Figure 3c), which is not clearly resolved in the DSC curve (Figure 3d). This comparison highlights the complementarity of the two techniques—while the instantaneous *S*(*Q*) difference reflects atomic-scale rearrangements, the DSC signal corresponds to the associated thermal effects.

The problem of identifying the crystalline products formed during the devitrification of the initially amorphous Mg_66_Zn_30_Ca_4_ alloy has been addressed in several scientific publications [17,18]. For example, Zhang et al. [17] described the crystallization sequence of this metallic glass as follows:−During the first phase transformation, the binary orthorhombic phase Mg_51_Zn_20_ crystallizes from the amorphous precursor. This phase belongs to space group Immm, with lattice parameters *a* = 14.224 Å, *b* = 14.195 Å, and *c* = 14.667 Å. A significant fraction of the amorphous matrix remains untransformed at this stage.−The second transformation involves the crystallization of the residual amorphous phase into the hexagonal Mg phase (P6_3_/mmc, *a* = 3.205 Å, *c* = 5.210 Å) and an intermetallic phase denoted as IM1, also with a hexagonal structure (P6_3_/mmc, *a* = 9.459 Å, *c* = 9.926 Å). This IM1 phase was identified by Schäublin et al. [18] as a solid solution with the general formula Ca_3_Mg_x_Zn_15−x_, with 4.6 ≤ *x* ≤ 12.−The third transformation leads to the coexistence of the Mg phase, the IM1 phase, and a new intermetallic phase designated as IM4. According to Schäublin et al., the chemical composition of IM4 is approximately Ca_1 5_Mg_55 3_Zn_43 2_; however, its crystal structure has not yet been resolved.−The fourth transformation results in the formation of Mg, IM1, and yet another new intermetallic phase, Ca_8_Mg_26 1_Zn_57 9_ IM3. Similar to the IM1, the IM3 phase exhibits a hexagonal lattice (P6_3_/mmc) with lattice parameters *a* = 14.748 Å and *c* = 8.783 Å.

As this overview shows, the crystallization of Mg_66_Zn_30_Ca_4_ metallic glass is highly complex and remains only partially understood. In the present study, we aimed to confirm the previously proposed phase sequence through in situ experiments using monochromatic high-energy X-ray radiation. Based on our measurements, we were able to confidently identify all products of the first and fourth crystallization events. However, despite the use of high-resolution data, we were not able to fully resolve the phase composition after the second and third transformations.

### 3.4. First Crystallization

In the temperature range from 420 K to 460 K, the originally fully amorphous Mg_66_Zn_30_Ca_4_ alloy undergoes partial crystallization, forming the binary orthorhombic phase Mg_51_Zn_20_, while the remainder of the alloy remains in a residual amorphous state. This temperature interval is highlighted in orange in Figure 4a in the DSC plot.

The corresponding XRD pattern acquired at 433 K is shown in Figure 4b. In this figure, the experimental data are represented by blue crosses, the Rietveld refinement fit by the green curve, and the red line denotes the background contribution associated with the residual amorphous matrix. The crystalline phase refined in the pattern is the orthorhombic Mg_51_Zn_20_ phase with space group Immm (No. 71), and its atomic arrangement within the unit cell is illustrated in Figure 4d.

The evolution of the lattice parameters of this phase with temperature is shown in Figure 4c. A linear regression analysis of the data reveals that the greatest thermal expansion occurs along the c-axis, which increases by approximately 0.21% over this 40 K range. In comparison, the b-parameter increases by 0.14%, and the a-parameter by only 0.01%. These results clearly indicate that the Mg_51_Zn_20_ phase exhibits anisotropic thermal expansion, which is a reference characteristic in crystallographic terms, particularly along the Z-direction.

### 3.5. Fourth Crystallization

In the narrow temperature range between 614 K and 625 K, just prior to the melting of the alloy (see Figure 5a), two crystalline phases are present: the hexagonal intermetallic phase *IM3* with chemical composition Ca_8_Mg_26 1_Zn_57 9_ and space group P6_3_/mmc, and a simple hexagonal phase corresponding to nearly pure magnesium. The Rietveld refinement of the structural models for both phases is shown in Figure 5b, while their thermal expansion behavior is presented in Figure 5c,d. The schematic representations of their respective unit cell contents are displayed in Figure 5e (*IM3*) and 5f (Mg).

In contrast to the findings reported in article [17], we have refined the phase composition of the alloy in its final crystallized state before melting, and our results show that the intermetallic phase *IM1* Ca_3_Mg_x_Zn_15−x_, with 4.6 ≤ *x* ≤ 12, is not present at this stage.

## 4. Conclusions

In this study, the structural relaxation and complex crystallization behavior of Mg_66_Zn_30_Ca_4_ metallic glass were investigated using in situ high-energy synchrotron X-ray diffraction and differential scanning calorimetry. The aim was to clarify phase formation pathways during thermal treatment and to verify previously proposed crystallization sequences.

Mg_66_Zn_30_Ca_4_ metallic glass shows a distinct supercooled liquid region of 58 K, enabling thermoplastic processing before crystallization.Structural relaxation is nearly complete in the first heating cycle, and repeated heating–cooling cycles up to 396 K induce no signs of devitrification.Crystallization proceeds in four distinct steps. During the first crystallization stage, the alloy partially transforms into the orthorhombic Mg_51_Zn_20_ phase, while a significant fraction of the amorphous matrix remains untransformed. In the final (fourth) crystallization stage, two crystalline phases were identified: nearly pure hexagonal Mg and the intermetallic Ca_8_Mg_26 1_Zn_57 9_ phase (*IM3*). The presence of the hexagonal Ca_3_Mg_x_Zn_15−x_ phase (*IM1*, with 4.6 ≤ x ≤ 12), which was reported in a previous study [17,18] to appear during the last stage of crystallization, was not confirmed in our measurements.The orthorhombic Mg_51_Zn_20_ phase exhibits anisotropic thermal expansion, with the highest relative increase along the c-axis.The combination of in situ synchrotron X-ray diffraction and Rietveld refinement is effective for identifying phase evolution and lattice parameter changes in multicomponent metallic glasses.

## Figures and Tables

**Figure 1 materials-18-02464-f001:**
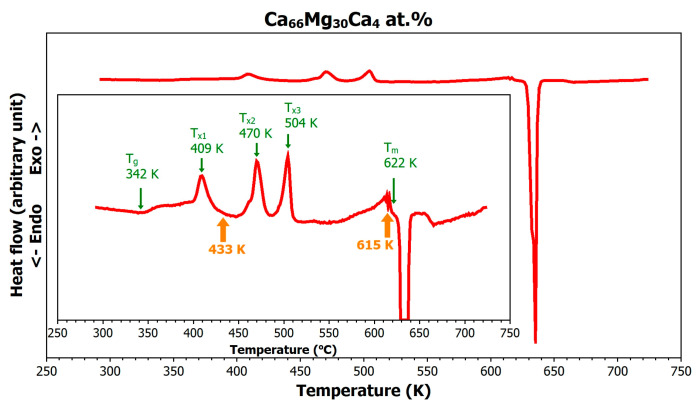
DSC curve of the initially amorphous Mg_66_Zn_30_Ca_4_ alloy obtained at a heating rate of 5 K·min^−1^. Orange arrows mark temperatures corresponding to X-ray measurements shown later in Figures 4b and 5b.

**Figure 2 materials-18-02464-f002:**
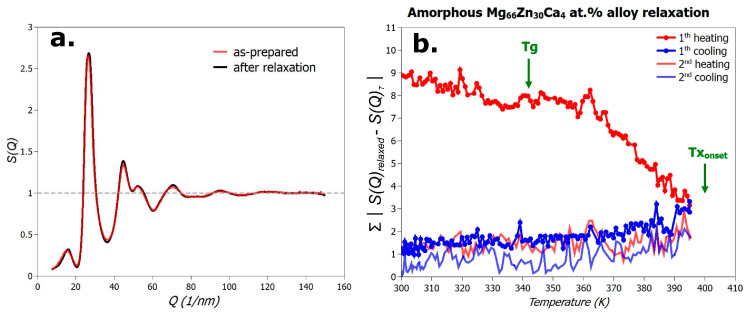
(**a**) Comparison of X-ray structure factors, *S*(*Q*), for the Mg_66_Zn_30_Ca_4_ alloy in the as-prepared (red curve) and relaxed state after two heating–cooling cycles (black curve). (**b**) Quantitative difference between structure factors relative to the fully relaxed state.

**Figure 3 materials-18-02464-f003:**
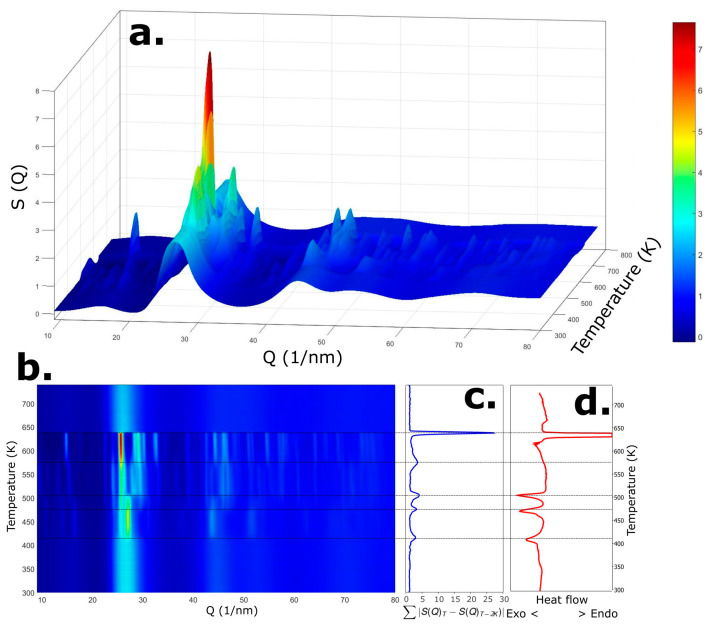
(**a**) 3D plot showing the evolution of the structure factor, *S*(*Q*), of the Mg_66_Zn_30_Ca_4_ alloy as a function of annealing temperature. (**b**) Top-down view of the same dataset. (**c**) Instantaneous change of the structure factor, as defined by the area difference between adjacent measurements. (**d**) DSC trace shown for comparison with c.

**Figure 4 materials-18-02464-f004:**
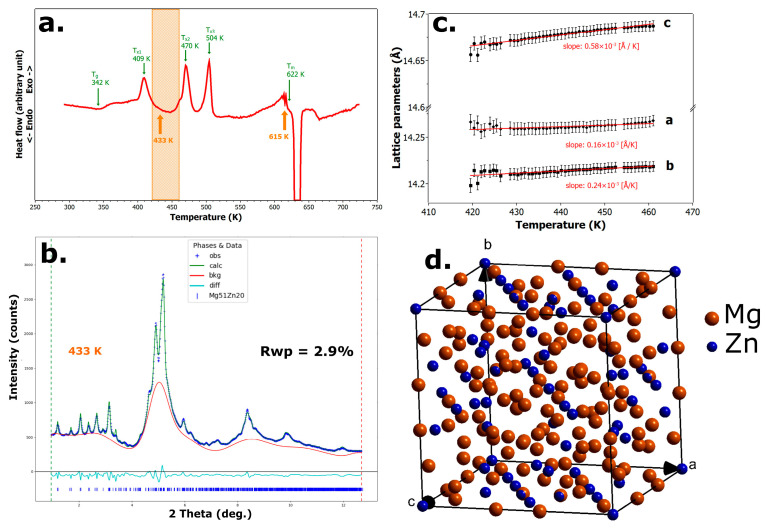
The first crystallization of the Mg_66_Zn_30_Ca_4_ alloy. (**a**) DSC curve with the analyzed temperature range highlighted in orange. (**b**) Rietveld refinement of the alloy annealed at 433 K. (**c**) Temperature dependence of the lattice parameters of the Mg_51_Zn_20_ phase. (**d**) Schematic representation of the unit cell content of the Mg_51_Zn_20_ phase.

**Figure 5 materials-18-02464-f005:**
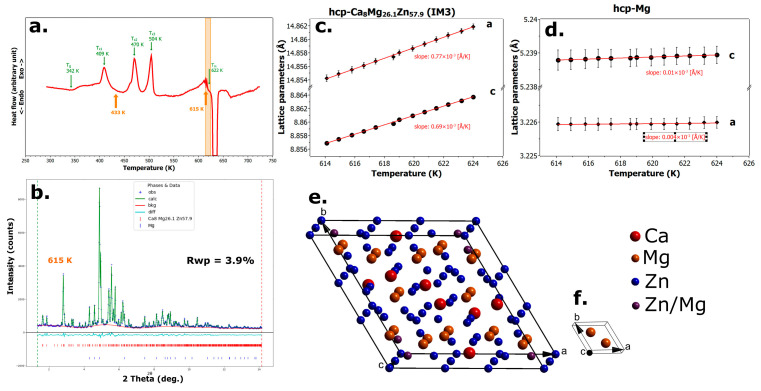
The fourth crystallization of the Mg_66_Zn_30_Ca_4_ alloy. (**a**) DSC curve with the analyzed temperature range highlighted in orange. (**b**) Rietveld refinement of the alloy annealed at 615 K. (**c**) Temperature dependence of the lattice parameters of the *IM3* phase. (**d**) Temperature dependence of the lattice parameters of the Mg phase. (**e**) Schematic illustration of the unit cell content of the *IM3* phase. (**f**) Schematic illustration of the unit cell content of the Mg phase.

## Data Availability

The original contributions presented in this study are included in the article. Further inquiries can be directed to the corresponding author(s).

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
