# Peer review of "Relaxation and Devitrification of Mg66Zn30Ca4 Metallic Glass"

_materials, 2025, doi:10.3390/ma18112464_

Round 1

Reviewer 1 Report

Comments and Suggestions for Authors

1, In Section 3.2, the authors state that the relaxed reference state was chosen to be S(Q) recorded at 300 K after the second cooling cycle. Why was this state selected as the reference? Please interpret it.

2., In Section 3.4, the thermal expansion along the a-axis is only 0.01%. Was the instrumental error excluded?

3, In Figure 4(b), the Rietveld refinement fit is shown by the green curve, Did the refinement process consider potential contributions from the residual amorphous phase?

Author Response

We thank Reviewer 1 for his/her constructive comments and valuable suggestions, which have significantly contributed to improving the clarity and quality of our manuscript.

 1., In Section 3.2, the authors state that the relaxed reference state was chosen to be S(Q) recorded at 300 K after the second cooling cycle. Why was this state selected as the reference? Please interpret it.

Thank you for this valuable comment. The relaxed reference state was specifically chosen as the structure factor S(Q) recorded at 300 K after the second cooling cycle because this state represents the most energetically stable and structurally equilibrated condition achievable under the experimental conditions used in our study. After the first thermal cycle, structural relaxation was nearly complete, as evidenced by minimal changes upon subsequent thermal cycling. Thus, selecting this second-cycle room-temperature measurement ensured that our reference structure represented a fully relaxed amorphous state, allowing accurate quantification of structural changes relative to an equilibrium reference.

This assumption is further supported by the fact that the integrated difference in the structure factors for all other measurements is greater than that after the second cooling to room temperature.

2., In Section 3.4, the thermal expansion along the a-axis is only 0.01%. Was the instrumental error excluded?

As can be seen from Figure 4c, the refined lattice parameters of the Mg₅₁Zn₂₀ phase are displayed together with their estimated errors. Indeed, the variation of the parameter a within the temperature interval of 420–460 K falls within this error margin. However, the changes in the other lattice parameters, especially parameter c, are significantly larger, which confirms our statement that the thermal expansion of this phase is anisotropic.

3., In Figure 4(b), the Rietveld refinement fit is shown by the green curve, Did the refinement process consider potential contributions from the residual amorphous phase?

Thank you for highlighting this important aspect. Indeed, the Rietveld refinement explicitly accounted for the contribution of the residual amorphous phase. This contribution is illustrated by the red line in Figure 4(b), representing the fitted amorphous background. The refinement process thus considered both the crystalline diffraction peaks (fitted by the green curve) and the underlying amorphous phase. This approach ensured accurate modeling of the phase fractions and crystalline parameters while properly isolating the crystalline signal from the amorphous background.

Reviewer 2 Report

Comments and Suggestions for Authors

The authors investigated the structural relaxation and crystallization behavior of one Mg-Zn-Ca MG through high-energy synchrotron-based X-ray diffraction. The following issues should be addressed before publication:

- Abstract should be revised, the authors defined IM3 phase but never mentioned this phase again in abstract, and did not explain which phase of IM4. The abstract can stand beyond the main text alone, thus these abbreviations should be carefully used.

- The sentences of “The glass transition temperature of the alloy is observed at 342 K, and the onset of the first crystallization occurs at 400 K. The temperature interval between these two points (38 K)” is confusing. If the Tg-onset and Tg-end were considered when determining the width of SCL region, this should be clearly explained in the text and illustrated in Figure 1.

- These abbreviations should be defined consistently as they appeared in the text. IM4 now appeared before IM3, and IM2 was completely missed in current version.

-The difference concerning the crystallization phases between this study and previous studies should be interpreted. The reason of the absence of IM4 is not well explained.

- The reference in line 223 was missed.

Author Response

We sincerely thank Reviewer 2 for his/her insightful comments and constructive suggestions. Below, we address each point and indicate how the manuscript has been revised accordingly.

The authors investigated the structural relaxation and crystallization behavior of one Mg-Zn-Ca MG through high-energy synchrotron-based X-ray diffraction. The following issues should be addressed before publication:

- Abstract should be revised, the authors defined IM3 phase but never mentioned this phase again in abstract, and did not explain which phase of IM4. The abstract can stand beyond the main text alone, thus these abbreviations should be carefully used.

Thank you for pointing this out. We have revised the abstract to clearly mention all relevant phases. All changes in the revised manuscript text are in red.

- The sentences of “The glass transition temperature of the alloy is observed at 342 K, and the onset of the first crystallization occurs at 400 K. The temperature interval between these two points (38 K)” is confusing. If the Tg-onset and Tg-end were considered when determining the width of SCL region, this should be clearly explained in the text and illustrated in Figure 1.

We appreciate the reviewer’s comment regarding clarity. The width of the supercooled liquid region (ΔT) in our manuscript is defined as the temperature interval between the glass transition temperature (Tgonset at 342 K) and the onset of the first crystallization event (Tx1 at 400 K). To clarify this point, the manuscript text has been revised, explicitly stating how ΔT was determined.

- These abbreviations should be defined consistently as they appeared in the text. IM4 now appeared before IM3, and IM2 was completely missed in current version.

The intermetallic phases IM1 (CaMgₓZn₁₅₋ₓ, with 4.6 ≤ x ≤ 12), IM3 (CaMg₂₆.Zn₅₇.), and IM4 (Ca.Mg₅₅.Zn₄₃.) are now well-established in the scientific literature [19,20]. Their order of appearance in the manuscript corresponds to their crystallization sequence from the Mg₆₆Zn₃₀Ca metallic glass upon increasing temperature. The presence of the intermetallic phase IM2 (Ca₁₄.Mg₁₅.Zn₆₉.) was not confirmed in the scientific study by Zhang et al. [19].

-The difference concerning the crystallization phases between this study and previous studies should be interpreted. The reason of the absence of IM4 is not well explained.

In our study, unlike the work by Zhang et al., [19] we used high-energy synchrotron monochromatic radiation and the experimental setup that enables truly precise identification of the phase composition. Our results show that, just prior to melting, the alloy consists exclusively of two phases: IM3 and Mg, as confirmed by the Rietveld fit of the measured data. In contrast, study [19] employed a laboratory diffractometer with insufficient resolution. Moreover, unlike our experiment, the measurements by Zhang et al. [19] were conducted ex-situ, after cooling from 593 K. We have completely reformulated point 3 in the Conclusions section. The revised text is highlighted in red.

- The reference in line 223 was missed.

Thank you for noticing this oversight. We add the reference

Reviewer 3 Report

Comments and Suggestions for Authors

Relaxation and devitrification of metallic glasses are typical topics and important. The current work provides pretty insights and deserves publication. The current content needs to be improved. Please answer the following questions in detail.

  1. When studying the thermal stability of the Mg66Zn30Ca4 metallic glass, although multiple phase transition temperatures are mentioned, the microstructural factors influencing these transition temperatures are not deeply explored. For example, how do the structure and distribution of atomic clusters in the alloy affect the glass transition temperature and the onset temperature of crystallization? Can a more in - depth analysis be carried out from the atomic level?
  2. The article indicates that the structural relaxation is nearly complete during the first heating cycle, but the long - term impact of this relaxation process on the subsequent properties of the alloy, such as mechanical properties and corrosion resistance, is not studied. How does structural relaxation affect the performance stability of the alloy in practical applications? Will the change in the relaxation state after multiple thermal cycles affect the alloy's performance?
  3. In the crystallization process of the Mg66Zn30Ca4 metallic glass, the phase composition after the second and third transformations could not be fully resolved. Are there any potential methods, such as using other advanced characterization techniques or adjusting the experimental conditions, that might help to clarify the phase composition more accurately?
  4. The paper mentions the anisotropic thermal expansion of the orthorhombic Mg51Zn20 phase. How does this anisotropic thermal expansion behavior affect the overall dimensional stability and mechanical integrity of the alloy during thermal cycling or in service conditions?
  5. The study focuses on the relaxation and devitrification of the Mg66Zn30Ca4 metallic glass. However, it does not discuss how the processing parameters during the preparation of the amorphous ribbons, like the melt - spinning speed, affect the initial structure and subsequent properties of the metallic glass. How do these preparation parameters interact with the thermal behavior studied in the paper?

Author Response

We sincerely thank Reviewer 3 for his/her thoughtful and insightful feedback. We address each point in detail below and suggest modifications or clarifications in the manuscript where appropriate.

Relaxation and devitrification of metallic glasses are typical topics and important. The current work provides pretty insights and deserves publication. The current content needs to be improved. Please answer the following questions in detail.

1. When studying the thermal stability of the Mg66Zn30Ca4 metallic glass, although multiple phase transition temperatures are mentioned, the microstructural factors influencing these transition temperatures are not deeply explored. For example, how do the structure and distribution of atomic clusters in the alloy affect the glass transition temperature and the onset temperature of crystallization? Can a more in - depth analysis be carried out from the atomic level?

Thank you for this excellent suggestion. In our previous work [18], we investigated the atomic structure of Mg₆₆Zn₃₀Ca and identified a high fraction of icosahedrally packed clusters, which contribute significantly to the alloy’s thermal stability. These clusters impede atomic diffusion, raise the glass transition temperature (Tg), and delay the onset of crystallization (Tx) by stabilizing the amorphous network.

To address the reviewer’s comment, we have added a brief explanation to the end of Section 3.1, highlighting the connection between atomic clustering and the observed thermal transitions.

2. The article indicates that the structural relaxation is nearly complete during the first heating cycle, but the long - term impact of this relaxation process on the subsequent properties of the alloy, such as mechanical properties and corrosion resistance, is not studied. How does structural relaxation affect the performance stability of the alloy in practical applications? Will the change in the relaxation state after multiple thermal cycles affect the alloy's performance?

The structural relaxation may significantly influence mechanical behavior and corrosion resistance, especially in applications where thermal cycling occurs. While these properties were beyond the scope of the present study, prior investigations on similar Mg-based glasses suggest that relaxation tends to reduce internal stresses, improve strength uniformity, and stabilize electrochemical degradation rates.

By studying the relaxation behavior, we also evaluated the stability of the alloy within the supercooled liquid region, where the alloy remains amorphous even during the second heating up to the onset temperature of the first crystallization (Tx onset).

3. In the crystallization process of the Mg66Zn30Ca4 metallic glass, the phase composition after the second and third transformations could not be fully resolved. Are there any potential methods, such as using other advanced characterization techniques or adjusting the experimental conditions, that might help to clarify the phase composition more accurately?

We appreciate this important point. As the reviewer noted, the identification of phases during the intermediate crystallization steps remains challenging due to overlapping diffraction peaks and possible nanocrystalline domains. To improve resolution in future work, we are considering complementary techniques such as:

High-resolution transmission electron microscopy (HRTEM) for direct imaging of nanophases,

High-resolution XRD data combined with neutron diffraction ND for better contrast between light and heavy elements.

4. The paper mentions the anisotropic thermal expansion of the orthorhombic Mg51Zn20 phase. How does this anisotropic thermal expansion behavior affect the overall dimensional stability and mechanical integrity of the alloy during thermal cycling or in service conditions?

The anisotropic thermal expansion of the Mg₅₁Zn₂₀—especially the larger expansion along the c-axis—may induce internal stresses during thermal cycling, particularly when embedded in a composite microstructure with coexisting amorphous and crystalline regions. Such stress development can influence crack formation or deformation behavior under crystallisation.

5. The study focuses on the relaxation and devitrification of the Mg66Zn30Ca4 metallic glass. However, it does not discuss how the processing parameters during the preparation of the amorphous ribbons, like the melt - spinning speed, affect the initial structure and subsequent properties of the metallic glass. How do these preparation parameters interact with the thermal behavior studied in the paper?

We thank the reviewer for this important observation. Indeed, the processing parameters during melt-spinning—such as the melt temperature, crucible material and geometry, and the rotation speed of the copper wheel—have a significant influence on the cooling rate and thus the resulting atomic structure of the amorphous ribbons.

Our research group has been engaged in the preparation of metallic glass ribbons for over ten years. During this period, we have conducted numerous experimental trials to optimize processing parameters for different alloy systems. The current process settings for Mg₆₆Zn₃₀Ca₄ were established based on this extensive experience to achieve a fully amorphous structure with consistent thermal and structural behavior.

However, a systematic study comparing how different preparation parameters affect the structural relaxation and devitrification behavior of the Mg₆₆Zn₃₀Ca₄ alloy would require a dedicated research effort. Such a study would entail producing a large set of ribbons under varied processing conditions, followed by comprehensive characterization—ideally through in-situ synchrotron X-ray diffraction. This, in turn, would demand substantial beamtime at a high-brilliance synchrotron facility such as PETRA III, which is currently beyond the scope of this work.

Reviewer 4 Report

Comments and Suggestions for Authors

 In this study, the authors investigated the Mg₆₆Zn₃₀Ca₄ metallic glass, a promising biodegradable material due to its high strength, corrosion resistance, and excellent glass-forming ability. They employed high-energy synchrotron-based X-ray diffraction and differential scanning calorimetry (DSC) to examine the thermal stability, structural relaxation, and crystallization behavior of the as-spun Mg₆₆Zn₃₀Ca₄ alloy. The results reveal a four-step crystallization process, and the authors successfully elucidate the first and fourth steps in terms of the resulting crystalline phases. This paper provides detailed insights into the as-spun Mg₆₆Zn₃₀Ca₄ alloy and is clearly of interest. I recommend its consideration for publication in Materials after the following minor comments are addressed.

Comment 1: 

             In general, Mg and Zn are prone to evaporation during solidification. Therefore, I recommend the authors provide the actual overall composition of the Mg₆₆Zn30Ca4 alloy, as measured by inductively coupled plasma (ICP) analysis.

Comment 2: 

             As I understand it, the glass transition temperature (Tg) is typically observed as the onset of an endothermic heat flow, since the absorbed heat enables atomic movement within a short range, accompanied by a change in viscosity. However, in Figure 1, the authors indicate an exothermic temperature as Tg. Please double-check this assignment and provide an explanation. (see Figure1 in https://www.nature.com/articles/ncomms14679)

Comment 3: 

According to the literature cited in the manuscript, the IM4 phase typically forms after the third crystallization step. Therefore, it is unlikely to be present after the fourth crystallization, as it may transform into other phases during this stage. Since the current study examines the alloy after the fourth crystallization step, the absence of the IM4 phase is expected. Please address and discuss this point in the manuscript.

Comment 4: 

Due to the limited penetration depth of X-rays, high-energy synchrotron-based X-ray diffraction may provide structural information that is more representative of the surface. To complement this, I suggest including cross-sectional TEM observations after the first and fourth crystallization steps. Additionally, please expand the manuscript by adding dedicated sections—3.4 First Crystallization and 3.5 Fourth Crystallization—with corresponding discussions.

Author Response

We sincerely thank Reviewer 4 for their positive evaluation and constructive suggestions. Below, we provide detailed responses and describe how we revised the manuscript.

In this study, the authors investigated the Mg₆₆Zn₃₀Ca₄ metallic glass, a promising biodegradable material due to its high strength, corrosion resistance, and excellent glass-forming ability. They employed high-energy synchrotron-based X-ray diffraction and differential scanning calorimetry (DSC) to examine the thermal stability, structural relaxation, and crystallization behavior of the as-spun Mg₆₆Zn₃₀Ca₄ alloy. The results reveal a four-step crystallization process, and the authors successfully elucidate the first and fourth steps in terms of the resulting crystalline phases. This paper provides detailed insights into the as-spun Mg₆₆Zn₃₀Ca₄ alloy and is clearly of interest. I recommend its consideration for publication in Materials after the following minor comments are addressed.

Comment 1: 

             In general, Mg and Zn are prone to evaporation during solidification. Therefore, I recommend the authors provide the actual overall composition of the Mg₆₆Zn30Ca4 alloy, as measured by inductively coupled plasma (ICP) analysis.

We thank the reviewer for this comment. Instead of ICP, the chemical composition of the as-spun amorphous ribbon was verified using energy-dispersive X-ray spectroscopy (EDX) in a scanning electron microscope (SEM), which provides sufficient accuracy for this type of alloy system. The measured composition was:

Mg = 65.7 at.%, Zn = 29.8 at.%, Ca = 4.5 at.%,

which is in very good agreement with the nominal composition Mg₆₆Zn₃₀Ca, confirming minimal elemental loss during processing. This information has now been added to Section 2 (Materials and Methods).

Comment 2: 

             As I understand it, the glass transition temperature (Tg) is typically observed as the onset of an endothermic heat flow, since the absorbed heat enables atomic movement within a short range, accompanied by a change in viscosity. However, in Figure 1, the authors indicate an exothermic temperature as Tg. Please double-check this assignment and provide an explanation. (see Figure1 in https://www.nature.com/articles/ncomms14679)

In our DSC measurement, the glass transition (Tg) was identified as the onset of a subtle baseline deviation prior to the first exothermic crystallization peak, which is consistent with established methodologies. The Tg is endothermic event marked by the green arrow, however, due to the low signal intensity it may appear ambiguous.

Comment 3: 

According to the literature cited in the manuscript, the IM4 phase typically forms after the third crystallization step. Therefore, it is unlikely to be present after the fourth crystallization, as it may transform into other phases during this stage. Since the current study examines the alloy after the fourth crystallization step, the absence of the IM4 phase is expected. Please address and discuss this point in the manuscript.

In the original version of our manuscript, the IM4 phase was incorrectly mentioned. The correct statement is that the IM1 phase (CaMgₓZn₁₅₋ₓ, with 4.6 ≤ x ≤ 12) was not observed. This has been corrected in the revised version of the manuscript. We thank you very much for your attention to this detail.

Comment 4: 

Due to the limited penetration depth of X-rays, high-energy synchrotron-based X-ray diffraction may provide structural information that is more representative of the surface. To complement this, I suggest including cross-sectional TEM observations after the first and fourth crystallization steps. Additionally, please expand the manuscript by adding dedicated sections—3.4 First Crystallization and 3.5 Fourth Crystallization—with corresponding discussions.

We agree with the reviewer that cross-sectional techniques such as TEM can offer high-resolution insight into local structural features. However, the preparation of TEM specimens from this metallic glass after crystallization is extremely challenging due to the alloy’s high brittleness and tendency to fracture during thinning. Additionally, we believe that TEM analysis is not essential in this case, as we employed high-energy monochromatic synchrotron X-ray radiation, which offers deep penetration and provides structural information representative of both the surface and the bulk of the material.

Reviewer 5 Report

Comments and Suggestions for Authors

The submission is coherent, logically structured, and based on reliable experimental data. The authors provide a correct and detailed description of the structural relaxation and crystallization processes in the Mg₆₆Zn₃₀Ca₄ alloy, using DSC and in-situ synchrotron X-ray diffraction methods. The conclusions are directly supported by the experimental results and are appropriately interpreted. The terminology and methodology employed are appropriate. References to the literature are used correctly to compare the obtained results with previous studies. Overall, the manuscript is scientifically sound and well-documented. It addresses an interesting and relevant topic. The research is thoroughly conducted and interpreted, and the conclusions are fully justified. I recommend the article for publication in its current form, without any revisions.

Round 2

Reviewer 2 Report

Comments and Suggestions for Authors

The revision is satisfied.